

# Massive corrections to entanglement in minimal $E_8$ Toda field theory

**Olalla A. Castro-Alvaredo**

Department of Mathematics, City, University of London,
10 Northampton Square EC1V 0HB, UK

* o.castro-alvaredo@city.ac.uk

## Abstract

In this letter we study the exponentially decaying corrections to saturation of the second Rényi entropy of one interval of length $\ell$ in minimal $E_8$ Toda field theory. It has been known for some time that the entanglement entropy of a massive quantum field theory in 1+1 dimensions saturates to a constant value for $m_1 \ell \gg 1$ where $m_1$ is the mass of the lightest particle in the spectrum. Subsequently, results by Cardy, Castro-Alvaredo and Doyon have shown that there are exponentially decaying corrections to this behaviour which are characterized by Bessel functions with arguments proportional to $m_1 \ell$. For the von Neumann entropy the leading correction to saturation takes the precise universal form $-\frac{1}{8}K_0(2m_1\ell)$ whereas for the Rényi entropies leading corrections which are proportional to $K_0(m_1\ell)$ are expected. Recent numerical work by Pálmai for the second Rényi entropy of minimal $E_8$ Toda has identified next-to-leading order corrections which decay as $e^{-2m_1\ell}$ rather than the expected $e^{-m_1\ell}$. In this paper we investigate the origin of this result and show that it is incorrect. An exact form factor computation of correlators of branch point twist fields reveals that the leading corrections are proportional to $K_0(m_1\ell)$ as expected.



# 1  Introduction

## 1.1  Entanglement entropy

Measures of entanglement, such as the entanglement entropy (EE), have attracted much attention in recent years, particularly in the context of one-dimensional many body quantum systems (see e.g. review articles in the special issue [1]). Among such systems, those enjoying conformal invariance in the scaling limit are of particular interest as they provide a theoretical and universal description of critical phenomena. In their seminal work Calabrese and Cardy [2] used principles of Conformal Field Theory (CFT) to study the (EE) [3] of quantum critical systems. Their results generalised previous work [4], provided theoretical support for numerical observations in critical quantum spin chains [5] and highlighted the fact that the EE encodes universal information about quantum critical points, such as the central change of the corresponding CFT. This information may be extracted numerically in a very efficient way, typically by employing Density Matrix Renormalization Group methods [6], and this has provided one of the main motivations to investigate measures of entanglement in critical and near-critical systems. From a mathematical physics viewpoint (the one taken in this paper) the investigation of the EE is driven by interest in developing a better (if possible, analytical) understanding of the universal properties of the ground state of extended many body quantum systems.

The EE is a measure of the amount of quantum entanglement, in a pure quantum state, between the degrees of freedom associated to two sets of independent observables whose union is complete on the Hilbert space. In the present paper, the two sets of observables correspond to the local observables in two complementary connected regions, $A$ and $B$, of a 1+1-dimensional massive quantum field theory (QFT), and we will consider only the case where the quantum state is the ground state. Let $|\Psi\rangle$ be such a ground state. Consider a space bi-partition of the theory as sketched in Figure 1. Then the EE associated to region $A$ may be expressed as $S(\ell) = -\mathrm{Tr}(\rho_A \log \rho_A)$ where $\rho_A = \mathrm{Tr}_B(|\Psi\rangle\langle\Psi|)$ is the reduced density matrix associated to subsystem $A$ and $\ell$ is the subsystem's length.

The EE defined above is also known as von Neumann entropy. Alternative, related definitions of the entanglement entropy have been proposed which are also frequently studied. A set of popular measures is provided by the Rényi entropies which are defined as

$$S_n(\ell) = \frac{\log \mathrm{Tr}\rho_A^n}{1-n},\tag{1}$$

and have the property $\lim_{n\to 1} S_n(\ell) = S(\ell)$. In this paper we will consider the case $n = 2$ where $S_2(\ell) = -\log \mathrm{Tr}\rho_A^2$. We will refer to this quantity as the *second Rényi entropy*. We choose this particular value in order to compare with results obtained in [7] for the same quantity by a different method.

As mentioned earlier, at quantum critical points, the scaling limit of the EE has been widely studied in CFT [2, 4, 5, 8–10] and in lattice realizations of critical systems such as quantum spin chains [11–17] and lattice models [18–20]. In particular, the combination of a geometric description, Riemann uniformization techniques and standard expressions for CFT partition functions is very fruitful. Recently [21], this was generalized to non-unitary CFT and to the EE of excited states [22, 23], where general formulae were obtained using also such techniques. Near critical points, the scaling limit is instead described by massive quantum field theory (QFT), and geometric techniques relying on conformal mappings break down. So far the most powerful way of studying the EE in massive QFT is by using an approach based on local *branch-point twist fields* [24–26]. This approach is very fruitful and complete as it allows both for numerical and analytical computations. In this context, the second Rényi entropy may be defined as:

$$S_2(\ell) = -\log\left(\epsilon^{4\Delta_2}\langle\mathcal{T}(0)\tilde{\mathcal{T}}(\ell)\rangle_2\right), \tag{2}$$

where $\mathcal{T}$ and $\tilde{\mathcal{T}} := \mathcal{T}^\dagger$ are the branch point twist fields,

$$\Delta_n = \frac{c}{24}\left(n - \frac{1}{n}\right), \tag{3}$$

is their conformal dimension (at criticality) [2, 27, 28] as a function of the central charge $c$, and $\epsilon$ is a non-universal short-distance cut-off. The expression $\langle\mathcal{T}(0)\tilde{\mathcal{T}}(\ell)\rangle_2$ above denotes the two-point function in the ground state for $n = 2$. An important subtlety is that branch point twist fields are local fields in a new QFT which is constructed as $n$ non-interacting copies (in this case 2) of the original QFT. In this context, they are interpreted as symmetry fields associated to the cyclic permutation symmetry of the "replica" theory.

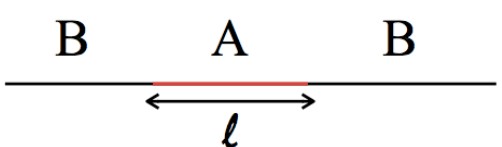

Figure 1: Typical bi-partition for the EE of one interval.

In this paper we aim to compare results based on a branch point twist field approach to recent numerical results by Tamás Pálmai [7] for the quantity (2). In [7] a new approach to the computation of the second Rényi entropy in massive integrable QFT was proposed which is based on the use of the Truncated Conformal Space Approach (TCSA) first proposed by Yurov and Zamolodchikov in [29]. The TCSA is based on Zamolodchikov's view of massive integrable models as massive perturbations of CFT [30]. It exploits the rich structure of the Hilbert space of CFT, perturbes and truncates the latter and then diagonalizes the "truncated" Hamiltonian. This provides a very successful way to access the low energy spectrum of massive integrable QFT with (a priori) any desired level of accuracy. The work [7] showed for the first time that TCSA can also be employed to access the quantity $\text{Tr}\rho_A^2$, and so may be applied to the study of measures of entanglement. This is a very interesting development which complements and enhances the existing twist field approach for massive QFTs.

## 1.2 The model

In this paper we consider an integrable massive QFT sometimes referred to as the critical Ising model in a magnetic field (IMMF) and also known as the minimal $E_8$ Toda field theory. In the spirit of Zamolodchikov's work [30], the theory can be described as a massive perturbation of the conformal Ising model whose operator content consists of simply three fields: the identity, the energy field $\varepsilon$ and the spin field $\sigma$. It is well-known that a massive perturbation by the

energy field gives rise to an integrable QFT known as the massive Ising model. This theory has a single particle and the two-body scattering matrix is simply $S(\theta) = -1$ as a function of the rapidity $\theta$. Surprisingly, perturbing with the spin field $\sigma$ instead gives rise to a much more complex but still integrable interacting QFT, the IMMF [30, 31]. The theory consists of 8 self-conjugate particles of different masses. All of the particles can also be formed as bound states of two other particles in the spectrum, that is, the corresponding two-body scattering matrices have a rich pole structure in the physical sheet with poles of up to order 12. Following Zamolodchikov's work, a plethora of papers by many authors led to the realization that the IMMF is but a particular case of a much wider family of integrable QFTs known as minimal Toda field theories (a detailed historical account of these findings can be found in [32, 35] and references therein). These in turn are "simplified" versions of another class of models, the Affine Toda field theories (ATFTs), in the sense that the $S$-matrices of ATFTs are equal those of minimal Toda theories, up to coupling-dependent multiplicative factors which have no poles in the physical sheet. ATFTs have been studied since a long time and have played a prominent role in the development of the field of integrable field theories [36, 37]. Based on the IMMF

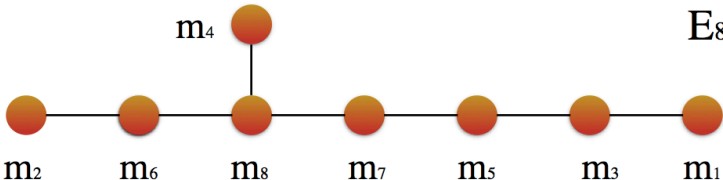

Figure 2: $E_8$ Dynkin diagram as a representation of the mass spectrum of the IMMF.

example and on extensive work on classical Toda theory it came to be expected that a different theory should exist for each simple Lie algebra and that this Lie algebraic structure would be crucial in the understanding of these models. Subsequently, a lot of work was carried out in order to compute the $S$-matrices of ATFTs related to each simple Lie algebra. Universal formulae for all simply-laced ATFTs were given in [32]. A universal description of the $S$-matrices of non-simply-laced ATFT based on $q$-deformed Coxeter elements was first proposed in [34] and further studied in [35] where a universal integral representation for all ATFT $S$-matrices was given. In this context, the eight particles in the spectrum of the IMMF are in one-to-one correspondence with the simple roots of $E_8$ (see Fig. 2). Indeed, their values are the Perron-Frobenius eigenvector of the corresponding Cartan matrix. A detailed account of the masses and scattering matrices of the theory can be found for instance in the review [38] and also in [39]. Here we will only report the data that we need for the present paper. We will only require the values of the masses of the four lightest particles in the spectrum

$$m_2 = 2m_1 \cos \frac{\pi}{5}, \quad m_3 = 2m_1 \cos \frac{\pi}{30} \quad \text{and} \quad m_4 = 2m_2 \cos \frac{7\pi}{30}. \tag{4}$$

where $m_1$ is the mass of the lightest particle. We note that the masses of the IMMF satisfy the inequality $m_i > 2m_1$ for $i > 3$ whereas $m_{1,2,3} < 2m_1$. In the following we will also require some of the two-particle scattering amplitudes as functions of the rapidity variable $\theta$. They generally have the structure:

$$S_{ab}(\theta) = \prod_{\alpha \in \mathscr{S}_{ab}} \left[ \frac{\tanh \frac{1}{2}(\theta + i\pi\alpha)}{\tanh \frac{1}{2}(\theta - i\pi\alpha)} \right]^{p_\alpha} \quad \forall \quad a, b = 1, \dots, 8. \tag{5}$$

where $\mathscr{S}_{ab}$ is a known set of integer values which characterises the scattering matrix and $p_\alpha$ are integer powers which determine the degeneracy of the poles of $S_{ab}(\theta)$ at $\theta = i\pi\alpha$. In

particular,

$$S_{11}(\theta) = \frac{\tanh\frac{1}{2}\left(\theta + \frac{2\pi i}{3}\right)}{\tanh\frac{1}{2}\left(\theta - \frac{2\pi i}{3}\right)} \frac{\tanh\frac{1}{2}\left(\theta + \frac{2\pi i}{5}\right)}{\tanh\frac{1}{2}\left(\theta - \frac{2\pi i}{5}\right)} \frac{\tanh\frac{1}{2}\left(\theta + \frac{i\pi}{15}\right)}{\tanh\frac{1}{2}\left(\theta - \frac{i\pi}{15}\right)}, \tag{6}$$

and

$$S_{12}(\theta) = \frac{\tanh\frac{1}{2}\left(\theta + \frac{4\pi i}{5}\right)}{\tanh\frac{1}{2}\left(\theta - \frac{4\pi i}{5}\right)} \frac{\tanh\frac{1}{2}\left(\theta + \frac{3\pi i}{5}\right)}{\tanh\frac{1}{2}\left(\theta - \frac{3\pi i}{5}\right)} \frac{\tanh\frac{1}{2}\left(\theta + \frac{7\pi i}{15}\right)}{\tanh\frac{1}{2}\left(\theta - \frac{7i\pi}{15}\right)} \frac{\tanh\frac{1}{2}\left(\theta + \frac{4\pi i}{15}\right)}{\tanh\frac{1}{2}\left(\theta - \frac{4i\pi}{15}\right)}, \tag{7}$$

are the only amplitudes we will require in this paper. As we can see, $S_{11}(\theta)$ has simple poles at $\frac{2\pi i}{3}, \frac{2\pi i}{5}$ and $\frac{\pi}{15}$ corresponding to the formation of particles 1, 2 and 3, respectively through the scattering processes $1 + 1 \to 1$, $1 + 1 \to 2$ and $1 + 1 \to 3$. Similarly, the scattering matrix $S_{12}(\theta)$ has four simple poles at $\frac{4\pi i}{5}, \frac{3\pi i}{5}, \frac{7i\pi}{15}$ and $\frac{4i\pi}{15}$ corresponding to the formation of bound states 1, 2, 3 and 4. Associated to these simple poles are the three-point couplings $\Gamma_{ab}^c$ defined as

$$i(\Gamma_{ab}^c)^2 = \text{Res}_{\theta = i\pi\alpha} S_{ab}(\theta). \tag{8}$$

In particular

$$(\Gamma_{11}^1)^2 = 2\sqrt{15 - 6\sqrt{5}}\cot\frac{\pi}{30}\cot^2\frac{2\pi}{15}, \quad (\Gamma_{11}^2)^2 = (\Gamma_{11}^1)^2\sqrt{9 + 4\sqrt{5}}\tan\frac{2\pi}{15}\tan\frac{7\pi}{30},$$

$$(\Gamma_{11}^3)^2 = (\Gamma_{11}^2)^2\frac{\tan\frac{\pi}{30}\tan\frac{\pi}{15}}{\sqrt{9 + 4\sqrt{5}}}, \quad \Gamma_{12}^1 = \Gamma_{11}^2, \quad (\Gamma_{12}^2)^2 = (\Gamma_{12}^1)^2(2 + \sqrt{5})\tan\frac{2\pi}{15}\cot^2\frac{\pi}{15}\cot\frac{7\pi}{30},$$

$$(\Gamma_{12}^3)^2 = (\Gamma_{12}^1)^2\cot\frac{\pi}{30}\cot\frac{\pi}{15}\cot\frac{2\pi}{15}\cot\frac{7\pi}{30}, \quad (\Gamma_{12}^4)^2 = (\Gamma_{12}^3)^2\tan\frac{\pi}{30}\tan\frac{2\pi}{15}. \tag{9}$$

will enter in some of the equations we will see later.

## 1.3  Structure of the paper

This paper is organised as follows: in section 2 we review the form factor approach for branch point twist fields. We propose expressions for some of the two-particle form factors in the IMMF as well as a set of consistency conditions that allow us to determine also the one-particle form factors of the four lightest particles in the spectrum. In section 3 we explain how the second Rényi entropy may be expressed in terms of twist field form factors and write down an expression including the six leading form factor corrections to its saturation value. We determine the precise coefficients of these corrections by solving the form factor equations proposed in section 2. We confirm the presence of exponentially decaying corrections, led by $e^{-m_1\ell}$. In section 4 we compare our results to those obtained in [7] by employing the TCSA approach and discuss their level of agreement. We present our conclusions and outlook in section 5.

# 2  Twist field form factors in the IMMF

## 2.1  Generalities

In 1+1 dimensional integrable QFT the most successful approach to computing multi-point functions of local operators is by expressing them in terms of form factors of individual fields. Form factors of local fields in integrable QFT can generally be computed exactly by pursuing the so-called form factor programme [40, 41] which was extended to the treatment of branch point twist fields in [24]. The programme has been carried out for countless models and fields and provides extremely accurate results for correlators, particularly two-point functions. Here

we are interested in the correlator (2) in the IMMF. Let $(a, j)$ represent the quantum numbers of a particle of type $a = 1, \ldots, 8$ living in copy $j = 1, \ldots, n$ (we will specialize to $n = 2$ later on). We may employ the so-called cumulant expansion [42–44]:

$$\log\left(\frac{\langle \mathcal{T}(0)\tilde{\mathcal{T}}(\ell)\rangle_n}{\langle \mathcal{T}\rangle_n^2}\right) = \sum_{k=1}^{\infty} \frac{c_k(\ell)}{k!(2\pi)^k}, \tag{10}$$

with

$$c_k(\ell) = \sum_{a_1,\ldots,a_k=1}^{8} \sum_{j_1,\ldots,j_k=1}^{2} \int_{-\infty}^{\infty} d\theta_1 \cdots \int_{-\infty}^{\infty} d\theta_k \, h_{j_1\ldots j_k}^{a_1\ldots a_k}(\theta_1, \cdots, \theta_k; n) e^{-\ell \sum_{i=1}^{k} m_{a_i} \cosh\theta_i}, \tag{11}$$

where the functions $h_{j_1\ldots j_k}^{a_1\ldots a_k}(\theta_1, \cdots, \theta_k; n)$ are given in terms of the form factors of the fields involved. In particular,

$$h_j^a(\theta; n) = |F_1^{\mathcal{T}|(a,j)}(\theta; n)|^2,$$
$$h_{j_1 j_2}^{a_1 a_2}(\theta_1, \theta_2; n) = F_2^{\mathcal{T}|(a_1,j_1)(a_2,j_2)}(\theta_1, \theta_2; n)(F_2^{\tilde{\mathcal{T}}|(a_1,j_1)(a_2,j_2)}(\theta_1, \theta_2; n))^* - h_{j_1}^{a_1}(\theta_1)h_{j_2}^{a_2}(\theta_2), \tag{12}$$

where

$$F_1^{\mathcal{T}|(a,j)}(\theta; n) := \frac{\langle 0|\mathcal{T}(0)|\theta\rangle_{(a,j)}}{\langle \mathcal{T}\rangle_n}, \quad F_2^{\mathcal{T}|(a_1,j_1)(a_2,j_2)}(\theta_1, \theta_2; n) := \frac{\langle 0|\mathcal{T}(0)|\theta_1\theta_2\rangle_{(a_1,j_1)(a_2,j_2)}}{\langle \mathcal{T}\rangle_n} \tag{13}$$

are the normalized one- and two-particle form factors. Here $\langle 0|$ represents the vacuum state and $|\theta\rangle_{(a,j)}$, $|\theta_1\theta_2\rangle_{(a_1,j_1)(a_2,j_2)}$ represent in-states of 1 and 2 particles, respectively. The states are characterized by the rapidities $\theta_i$ and particle quantum numbers $(a_i, j_i)$. In this paper we will only be interested in one and two particle form factor contributions which provide the most important contributions to (11) for large $\ell$. For spinless operators in relativistic theories we have that the one-particle form factors are rapidity independent. Therefore, from now on we will simply write $F_1^{\mathcal{T}|(a,j)}(\theta; n) := F_1^{\mathcal{T}|(a,j)}(n)$.

## 2.2 One- and two-particle form factors of the IMMF

Form factors of the IMMF where constructed in [39] for the stress-energy tensor (e.g. the spin field) and in [45] for the energy field. This construction can be easily adapted to the twist field by employing the programme proposed in [24]. In addition, here we only want to study a subset of the one- and two-particle form factors of the twist field which means that we do not need to engage into solving the complicated recursive equations that arise for higher particle numbers. We will also avoid the consideration of higher order poles which has been extensively discussed in [38, 39]. A particular feature of the computation of form factors of twist fields is that many of the basic formulae are very similar to those used in the construction of form factors of standard local fields, particularly when it comes to constructing the two-particle minimal form factor and therefore, what follows is strongly guided by the analysis of [39].

A minimal solution to the two-particle form factor equation will be denoted by

$$f_{a_1 a_2}(\theta_1 - \theta_2; n) := F_{\min}^{\mathcal{T}|(a_1,j)(a_2,j)}(\theta_1, \theta_2; n) \quad \text{with} \quad j = 1, \ldots, n. \tag{14}$$

This minimal form factor satisfies a twist field version of Watson's equations which may be summarised as [24]

$$f_{a_1 a_2}(\theta, n) = f_{a_1 a_2}(-\theta, n)S_{a_1 a_2}(\theta) = f_{a_1 a_2}(-\theta + 2\pi n i, n). \tag{15}$$

Let $S_{a_1 a_2}(\theta)$ have an integral representation of the form

$$S_{a_1 a_2}(\theta) = \exp\left[\int_0^\infty \frac{dt}{t} s_{a_1 a_2}(t) \sinh\left(\frac{t\theta}{i\pi}\right)\right], \tag{16}$$

where $s_{a_1 a_2}(\theta)$ is a function which depends of the theory and the particles $a_1, a_2$. Employing this integral representation it is easy to show that the solution to (15) may be written as

$$f_{a_1 a_2}(\theta; n) = \exp\left[\int_0^\infty \frac{dt}{t \sinh(nt)} s_{a_1 a_2}(t) \sin^2\left(\frac{it}{2}\left(n + \frac{i\theta}{\pi}\right)\right)\right]. \tag{17}$$

It is also well-known that if $S_{a_1 a_2}(\theta) = -1$ in (15) then the free Fermion solution $f_{a_1 a_2}(\theta) = -i \sinh\frac{\theta}{2n}$ is obtained. Combining these two results and comparing with the formulae provided in [39] for the minimal form factors of the IMMF we find

$$f_{a_1 a_2}(\theta; n) =$$
$$(-i \sinh\frac{\theta}{2n})^{\delta_{a_1 a_2}} \exp\left[\sum_{\alpha \in \mathscr{S}_{a_1 a_2}} 2p_\alpha \int_0^\infty dt \frac{\cosh\left((\alpha - \frac{1}{2})t\right)}{t \cosh\frac{t}{2} \sinh(nt)} \sin^2\left(\frac{it}{2}\left(n + \frac{i\theta}{\pi}\right)\right)\right], \tag{18}$$

where the exponential may be also expressed as the infinite product of gamma functions:

$$\prod_{\alpha \in \mathscr{S}_{a_1 a_2}} \prod_{k=0}^\infty \left[\frac{\Gamma\left(\frac{k+n-\alpha+1}{2n}\right)^2 \Gamma\left(\frac{k+n+\alpha}{2n}\right)^2}{\Gamma\left(\frac{k-\alpha-\frac{i\theta}{\pi}+1}{2n}\right) \Gamma\left(\frac{k+\alpha-\frac{i\theta}{\pi}}{2n}\right) \Gamma\left(1 + \frac{k-\alpha+\frac{i\theta}{\pi}+1}{2n}\right) \Gamma\left(1 + \frac{k+\alpha+\frac{i\theta}{\pi}}{2n}\right)}\right]^{p_\alpha(-1)^k} \tag{19}$$

and the set $\mathscr{S}_{a_1 a_2}$ has been defined after (5). Following [24] and [39], the most generic two-particle form factor takes the form

$$F_2^{\mathscr{T}|(a_1,j_1)(a_2,j_2)}(\theta_1, \theta_2; n) = \frac{Q_{a_1 a_2}^{j_1 j_2}(\theta_{12}; n)}{2n K_{j_1 j_2}(\theta_{12}; n)^{\delta_{a_1 a_2}} \prod_{\alpha \in \mathscr{S}_{a_1 a_2}} \left(B_\alpha(\theta_{12}; n)^{u(p_\alpha)} B_{1-\alpha}(\theta_{12}; n)^{v(p_\alpha)}\right)^{\delta_{j_1 j_2}}}$$
$$\times \frac{F_{\min}^{\mathscr{T}|(a_1,j_1)(a_2,j_2)}(\theta_1, \theta_2; n)}{F_{\min}^{\mathscr{T}|(a_1,j_1)(a_2,j_2)}(i\pi, 0; n)}, \tag{20}$$

with $\theta_{12} := \theta_1 - \theta_2$,

$$K_{j_1 j_2}(\theta; n) = \frac{\sinh\left(\frac{i\pi(1-2(j_1-j_2))-\theta}{2n}\right) \sinh\left(\frac{i\pi(1-2(j_1-j_2))+\theta}{2n}\right)}{\sin\frac{\pi}{n}}, \tag{21}$$

$$B_\alpha(\theta; n) = \sinh\left(\frac{i\pi\alpha - \theta}{2n}\right) \sinh\left(\frac{i\pi\alpha + \theta}{2n}\right), \tag{22}$$

and

$$u(2k+1) = k+1, \quad u(2k) = k \quad \text{and} \quad v(2k+1) = v(2k) = k \quad \text{for} \quad k \in \mathbb{Z}. \tag{23}$$

The function $K_{j_1 j_2}(\theta; n)$ encodes the full kinematic pole structure of the form factor, having kinematic poles at $\theta = i\pi$ and $\theta = i\pi(2n-1)$ in the extended physical strip $\text{Im}(\theta) \in [0, 2\pi n]$, whereas $B_\alpha(\theta; n)$ encodes the bound state pole structure, as characterised in [39]. The minimal form factors in (20) can be easily obtained from (18) by employing standard relations

which can be found for instance in [24]. Finally, the functions $Q_{a_1 a_2}^{j_1 j_2}(\theta; n)$ are solutions to the equations:

$$Q_{a_1 a_2}^{11}(\theta; n) = Q_{a_1 a_2}^{11}(-\theta; n) = Q_{a_1 a_2}^{11}(-\theta + 2\pi i n; n) \tag{24}$$

namely, they are linear combinations of functions of the type $\cosh^k \frac{\theta}{n}$ for $k = 1, 2, \ldots$, similar to the ansatz already employed in [39]. Which values of $k$ are involved will be determined by constraints on how $Q_{a_1 a_2}^{11}(\theta; n)$ behaves as $\theta \to \infty$ which we will discuss in the next subsection.

In this paper we will only require the two-particle form factors $F_2^{(1,1)(1,1)}(\theta_1, \theta_2; n)$ and $F_2^{(1,1)(2,1)}(\theta_1, \theta_2; n)$. These are special cases of (20) explicitly given by

$$F_2^{\mathcal{T}|(1,1)(1,1)}(\theta_1, \theta_2; n) = \frac{Q_{11}^{11}(\theta_{12}; n)}{2n K_{11}(\theta_{12}; n) \prod_{\alpha = \frac{2}{3}, \frac{2}{5}, \frac{1}{15}} B_\alpha(\theta_{12}; n)} \frac{f_{11}(\theta_{12}; n)}{f_{11}(i\pi; n)}, \tag{25}$$

and

$$F_2^{\mathcal{T}|(1,1)(2,1)}(\theta_1, \theta_2; n) = \frac{Q_{12}^{11}(\theta_{12}; n)}{2n \prod_{\alpha = \frac{4}{5}, \frac{3}{5}, \frac{7}{15}, \frac{4}{15}} B_\alpha(\theta_{12}; n)} \frac{f_{12}(\theta_{12}; n)}{f_{12}(i\pi; n)}, \tag{26}$$

with

$$f_{11}(\theta; n) = -i \sinh \frac{\theta}{2n} \exp\left[2 \int_0^\infty \frac{dt}{t} \frac{\cosh \frac{t}{10} + \cosh \frac{t}{6} + \cosh \frac{13t}{30}}{\cosh \frac{t}{2} \sinh(nt)} \sin^2\left(\frac{it}{2}\left(n + \frac{i\theta}{\pi}\right)\right)\right], \tag{27}$$

and

$$f_{12}(\theta; n) = \exp\left[2 \int_0^\infty \frac{dt}{t} \frac{\cosh \frac{t}{10} + \cosh \frac{3t}{10} + \cosh \frac{t}{30} + \cosh \frac{7t}{30}}{\cosh \frac{t}{2} \sinh(nt)} \sin^2\left(\frac{it}{2}\left(n + \frac{i\theta}{\pi}\right)\right)\right]. \tag{28}$$

## 2.3 Fixing one- and two-particle form factors

The equations (25)-(26) give the two-particle form factors of interest up to the functions $Q_{11}^{11}(\theta; n)$ and $Q_{12}^{11}(\theta; n)$. As anticipated earlier, these functions may be determined by employing additional constraints. In particular, the kinematic and bound state residue equations for the two-particle form factors require that:

$$\lim_{\bar{\theta}_0 \to \theta_0} (\theta_0 - \bar{\theta}_0) F_2^{\mathcal{T}|(a,j)(\bar{a},j)}(\bar{\theta}_0 + i\pi, \theta_0; n) = i, \tag{29}$$

and

$$\lim_{\bar{\theta}_0 \to \theta_0} (\theta_0 - \bar{\theta}_0) F_2^{\mathcal{T}|(a_1,j)(a_2,j)}\left(\bar{\theta}_0 + \frac{i\pi\alpha}{2}, \theta_0 - \frac{i\pi\alpha}{2}; n\right) = i \Gamma_{a_1 a_2}^{a_3} F_1^{\mathcal{T}|(a_3,j)}(n). \tag{30}$$

In the IMMF all particles are self conjugate so that the form factor (25) must satisfy the condition (29), giving the constraint:

$$Q_{11}^{11}(i\pi; n) = \prod_{\alpha = 5, 9, 14, 16, 21, 25, 30} \sin \frac{\pi\alpha}{30n}. \tag{31}$$

The same form factor possesses three bound state poles related to the formation of bound states, 1, 2 and 3. This means that it satisfies three versions of equation (30), giving three additional constraints:

$$Q_{11}^{11}\left(\frac{2\pi i}{3}; n\right) = -\Gamma_{11}^1 F_1^{\mathcal{T}|(1,1)}(n) \csc \frac{\pi}{n} \frac{f_{11}(i\pi; n)}{f_{11}(\frac{2\pi i}{3}; n)} \prod_{\alpha = 4, 5, 9, 11, 16, 20, 25} \sin \frac{\alpha\pi}{30n}, \tag{32}$$

$$Q_{11}^{11}(\frac{2\pi i}{5};n) = \Gamma_{11}^2 F_1^{\mathscr{T}|(2,1)}(n)\csc\frac{\pi}{n}\frac{f_{11}(i\pi;n)}{f_{11}(\frac{2\pi i}{5};n)}\prod_{\alpha=4,5,7,9,12,16,21}\sin\frac{\alpha\pi}{30n}, \tag{33}$$

$$Q_{11}^{11}(\frac{i\pi}{15};n) = -\Gamma_{11}^3 F_1^{\mathscr{T}|(3,1)}(n)\csc\frac{\pi}{n}\frac{f_{11}(i\pi;n)}{f_{11}(\frac{i\pi}{15};n)}\prod_{\alpha=2,5,7,9,11,14,16}\sin\frac{\alpha\pi}{30n}. \tag{34}$$

The form factor (26) has no kinematic poles but has four bound state poles associated to the formation of bound states 1, 2, 3 and 4. They give the additional constraints:

$$Q_{12}^{11}(\frac{4\pi i}{5};n) = \Gamma_{12}^1 F_1^{\mathscr{T}|(1,1)}(n)\frac{f_{12}(i\pi;n)}{f_{12}(\frac{4\pi i}{5};n)}\prod_{\alpha=3,5,8,16,19,21,24}\sin\frac{\alpha\pi}{30n}, \tag{35}$$

$$Q_{12}^{11}(\frac{3\pi i}{5};n) = -\Gamma_{12}^2 F_1^{\mathscr{T}|(2,1)}(n)\frac{f_{12}(i\pi;n)}{f_{12}(\frac{3\pi i}{5};n)}\prod_{\alpha=2,3,5,13,16,18,21}\sin\frac{\alpha\pi}{30n}, \tag{36}$$

$$Q_{12}^{11}(\frac{7i\pi}{15};n) = \Gamma_{12}^3 F_1^{\mathscr{T}|(3,1)}(n)\frac{f_{12}(i\pi;n)}{f_{12}(\frac{7\pi i}{15};n)}\prod_{\alpha=2,3,5,11,14,16,19}\sin\frac{\alpha\pi}{30n}, \tag{37}$$

and

$$Q_{12}^{11}(\frac{4i\pi}{15};n) = -\Gamma_{12}^4 F_1^{\mathscr{T}|(4,1)}(n)\frac{f_{12}(i\pi;n)}{f_{12}(\frac{4\pi i}{15};n)}\prod_{\alpha=3,5,8,11,13,16}\sin\frac{\alpha\pi}{30n}. \tag{38}$$

At this stage we have obtained 8 equations and have 4 unknowns, corresponding to the one particle form factors $F_1^{\mathscr{T}|(a,1)}(n)$ with $a = 1,2,3,4$ as well as the polynomials $Q_{11}^{11}(\theta;n)$ and $Q_{12}^{11}(\theta;n)$. We know from the two particle form factor equations that they must be even functions of $\theta$ and we would also like to require the cluster decomposition property which has been discussed in detail in [49] and observed for numerous models (a particularly rich example can be found in [50]), namely that

$$\lim_{\theta_1\to\infty} F_2^{\mathscr{T}|(1,1)(1,1)}(\theta_1,\theta_2;n) = (F_1^{\mathscr{T}|(1,1)}(n))^2. \tag{39}$$

and

$$\lim_{\theta_1\to\infty} F_2^{\mathscr{T}|(1,1)(2,1)}(\theta_1,\theta_2;n) = F_1^{\mathscr{T}|(1,1)}(n)F_1^{\mathscr{T}|(2,1)}(n). \tag{40}$$

These properties provide very strong constraints for the functions $Q_{11}^{11}(\theta;n)$ and $Q_{12}^{11}(\theta;n)$, as they allow us to determine the highest power of $e^{\theta/n}$ that can be involved. We have that

$$K_{11}(\theta;n) \sim -\frac{1}{4}e^{\frac{\theta}{n}} \quad\text{and}\quad B_\alpha(\theta;n) \sim -\frac{1}{4}e^{\frac{\theta}{n}} \quad\text{for}\quad \theta\to\infty. \tag{41}$$

It is also easy to determine the leading behaviours of $f_{11}(\theta;n)$ and $f_{12}(\theta;n)$ as $\theta\to\infty$. In this limit, integrals of the type

$$\exp\left[2\int_0^\infty dt\,\frac{\cosh\left((\alpha-\frac{1}{2})t\right)}{t\cosh\frac{t}{2}\sinh(nt)}\sin^2\left(\frac{it}{2}\left(n+\frac{i\theta}{\pi}\right)\right)\right], \tag{42}$$

may be approximated by changing variables to $x = t\theta$ and then expanding for small values of $\frac{x}{\theta}$. At leading and next-to-leading order, this yields the simple integral:

$$\exp\left[\int_0^\infty dx\left(\frac{2\theta}{nx^2}\sin^2\left(\frac{x}{2\pi}\right) - \frac{i}{x}\sin\left(\frac{x}{\pi}\right)\right)\right] = -ie^{\frac{\theta}{2n}}. \tag{43}$$

Numerical evaluation of the minimal form factors for $\theta$ large confirms the behaviour above, up to $n$-dependent proportionality constants which we, unfortunately, have been unable to find a convergent analytic expression for

$$f_{11}(\theta;n) \sim v(n)e^{\frac{2\theta}{n}}, \quad f_{12}(\theta;n) \sim u(n)e^{\frac{2\theta}{n}} \quad \text{for} \quad \theta \to \infty. \tag{44}$$

Nonetheless, the numbers $v(n)$ and $u(n)$ can be numerically estimated for every $n$. This means that, in order to satisfy (39)-(40) we need

$$Q_{11}^{11}(\theta;n) \sim e^{\frac{2\theta}{n}} \quad \text{and} \quad Q_{12}^{11}(\theta;n) \sim e^{\frac{2\theta}{n}} \quad \text{for} \quad \theta \to \infty. \tag{45}$$

thus, in general

$$Q_{11}^{11}(\theta;n) = A_{11}^{11}(n) + B_{11}^{11}(n)\cosh\frac{\theta}{n} + C_{11}^{11}(n)\cosh^2\frac{\theta}{n}, \tag{46}$$

and

$$Q_{12}^{11}(\theta;n) = A_{12}^{11}(n) + B_{12}^{11}(n)\cosh\frac{\theta}{n} + C_{12}^{11}(n)\cosh^2\frac{\theta}{n}. \tag{47}$$

And the cluster decomposition equations (39)-(40) can be written as

$$(F_1^{\mathscr{T}|(1,1)}(n))^2 = \frac{4^3\sin\frac{\pi}{n}C_{11}^{11}(n)v(n)}{2nf_{11}(i\pi;n)} \quad \text{and} \quad F_1^{\mathscr{T}|(2,1)}(n)F_1^{\mathscr{T}|(1,1)}(n) = \frac{4^3 C_{12}^{11}(n)u(n)}{2nf_{12}(i\pi;n)}. \tag{48}$$

It is worth noting that the same conclusions regarding the form of the functions $Q_{11}^{11}(\theta;n)$ and $Q_{12}^{11}(\theta;n)$ can be reached by appealing to a well-known argument presented for instance in [39] according to which the form factors of unitary operators (in particular, the twist field) can diverge at most as $e^{\Delta_n\theta_i}$ when one of the rapidities $\theta_i \to \infty$ and where $\Delta_n$ is the conformal dimension (3) of the twist field. In fact, for $c = \frac{1}{2}$ it turns out that $\Delta_n < 1$ for $n < 49$ and therefore, at least for a wide range of values of $n$ the form factors must tend to a constant as any of the rapidities they depend upon tends to infinity. The property of cluster decomposition, additionally establishes what this constant must be.

Putting together equations (31)-(38) and (48) we end up with 10 equations for 10 unknowns: $A_{11}^{11}(n), B_{11}^{11}(n), C_{11}^{11}(n), A_{12}^{11}(n), B_{12}^{11}(n), C_{12}^{11}(n)$ and the four one particle form factors $F_1^{\mathscr{T}|(1,1)}(n)$, $F_1^{\mathscr{T}|(2,1)}(n)$, $F_1^{\mathscr{T}|(3,1)}(n)$ and $F_1^{\mathscr{T}|(4,1)}(n)$. This means we are now in a position to determine all these functions and to investigate how our results apply to the study of the second Rényi entropy.

## 3 Second Rényi entropy of the IMMF

From the definition (2) and the expansion (10) we know that the Rényi entropy may be expressed as an infinite sum involving integrals over the form factors of the twist field. In particular the first few contributions can be written as

$$S_2(\ell) - S_2(\infty) = -\frac{2}{\pi}\sum_{a=1}^{8}|F_1^{\mathscr{T}|(a,1)}(2)|^2 K_0(m_a\ell)$$

$$-\frac{1}{(2\pi)^2}\sum_{a_1,a_2=1}^{8}\sum_{j=1}^{2}\int_{-\infty}^{\infty}d\theta_1\int_{-\infty}^{\infty}d\theta_2 h_{a_1a_2}^{1j}(\theta_1,\theta_2;2)e^{-m_{a_1}\ell\cosh\theta_1 - m_{a_2}\ell\cosh\theta_2} + \cdots \tag{49}$$

where

$$h_{a_1a_2}^{1j}(\theta_1,\theta_2;2) := |F_2^{\mathscr{T}|(a_1,1)(a_2,j)}(\theta_1,\theta_2;2)|^2 - |F_1^{\mathscr{T}|(a_1,1)}(2)|^2|F_1^{\mathscr{T}|(a_2,j)}(2)|^2. \tag{50}$$

and $S_2(\infty) = -4\Delta_2 \log(m\epsilon) - 2\log\langle\mathcal{T}\rangle_2$ is the non-universal saturation constant. The expansion above describes corrections to saturation which are exponentially decaying for large $m_1\ell$. If we consider all terms above, it becomes quickly apparent that some of the one-particle form factor contributions are subleading compared to some of the two-particle form factor contributions, for $\ell$ large. This is because, as mentioned earlier in the paper, the masses of particles $4, \ldots, 8$ are all larger than twice the mass of particle 1. In summary, this means that, the first six leading form-factor corrections to saturation in order of importance are

$$S_2(\ell) - S_2(\infty) = -\frac{2}{\pi}\sum_{a=1}^{3}|F_1^{\mathcal{T}|(a,1)}(2)|^2 K_0(m_a\ell)$$

$$-\frac{1}{2\pi^2}\sum_{j=1}^{2}\int_{-\infty}^{\infty}d\theta\, h_{11}^{1j}(\theta,0;2)K_0(2m_1\ell\cosh\theta/2) - \frac{2}{\pi}|F_1^{\mathcal{T}|(4,1)}(2)|^2 K_0(m_4\ell)$$

$$-\frac{1}{2\pi^2}\sum_{j=1}^{2}\int_{-\infty}^{\infty}d\theta\, h_{12}^{1j}(\theta,0;2)K_0(\ell\sqrt{m_1^2+m_2^2+2m_1m_2\cosh\theta}) - \cdots \quad (51)$$

where

$$\sum_{j=1}^{2} h_{1a}^{1j}(\theta,0;2) =$$
$$|F_2^{\mathcal{T}|(1,1)(a,1)}(\theta,0;2)|^2 + |F_2^{\mathcal{T}|(1,1)(a,1)}(-\theta+2\pi i,0;2)|^2 - 2|F_1^{\mathcal{T}|(1,1)}(2)|^2|F_1^{\mathcal{T}|(a,1)}(2)|^2, \quad (52)$$

and we have carried out one of the integrals in (49). For $m_1\ell \gg 1$ the leading contribution to the first integral in (51) can be written as:

$$\frac{1}{2\pi^2}\sum_{j=1}^{2}\int_{-\infty}^{\infty}d\theta\, h_{11}^{1j}(\theta,0;2)K_0(2m_1\ell\cosh\theta/2) \approx \sqrt{\frac{\pi}{m_1\ell}}\frac{e^{-2m_1\ell}}{4\pi^2}\sum_{j=1}^{2}\int_{-\infty}^{\infty}d\theta\,\frac{h_{11}^{1j}(\theta,0;2)}{\sqrt{\cosh\frac{\theta}{2}}}, \quad (53)$$

and similarly for the last integral

$$\frac{1}{2\pi^2}\sum_{j=1}^{2}\int_{-\infty}^{\infty}d\theta\, h_{12}^{1j}(\theta,0;2)K_0(\ell\sqrt{m_1^2+m_2^2+2m_1m_2\cosh\theta}) \approx$$
$$\sqrt{\frac{\pi}{2m_1\ell}}\frac{e^{-(m_1+m_2)\ell}}{2\pi^2}\sum_{j=1}^{2}\int_{-\infty}^{\infty}d\theta\,\frac{h_{12}^{1j}(\theta,0;2)}{\sqrt[4]{1+\frac{m_2^2}{m_1^2}+2\frac{m_2}{m_1}\cosh\theta}}, \quad (54)$$

where the remaining integrals are convergent and can be easily evaluated numerically.

## 3.1 Numerical computation of the one-particle form factors

Although we are now in a position to solve equations (31)-(38) and (39)-(40) and therefore obtain explicit formulae for the one- and two-particle form factors of interest, in practice these equations are rather cumbersome and finding exact formulae is extremely difficult (even for $n = 2$). They can however be very easily solved numerically for any given value of $n$.

An interesting observation from solving the equations numerically is that the solution is not unique. This is mainly due to the equations (39)-(40) which are quadratic in the one-particle form factors. There are in fact three solutions for each value of $n$. This obviously poses the question as to which of these solutions is the correct one. It also indicates that the form factor equations allow for several twist field solutions. This is not surprising as all operators enjoying

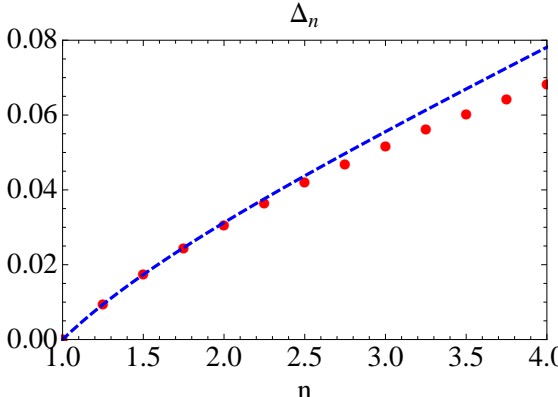

Figure 3: The function $\Delta_n = \frac{1}{48}\left(n - \frac{1}{n}\right)$ (dashed line) compared to values of $\Delta_n$ obtained by evaluating the $\Delta$-sum rule (circles) including the first 6 leading contributions to the form factor expansion (similar to (51)).

the twist-property should have form factors satisfying the same set of equations. Among those operators there is the branch point twist field we are interested in, but also other twist fields, such as the composite twist fields introduced in [46] and studied in [21, 47, 48]. These are fields which are defined (at criticality) as the leading field in the OPE of the standard branch point twist field and any other local field of the replica theory.

Fortunately, there is one simple way of telling such composite twist fields and the twist fields we are interested in apart: composite twist fields have form factors which do not vanish at $n = 1$ whereas the twist field $\mathscr{T}$ must reduce to the identity field at $n = 1$, hence all its form factors vanish at $n = 1$. Imposing this condition we find a single solution with the desired property of having vanishing form factors at $n = 1$. In addition, there are certain consistency checks that we may further apply to test this solution. A common test is the $\Delta$-sum rule proposed in [49] which may be written as

$$\Delta_n = -\frac{1}{2\langle\mathscr{T}\rangle_n}\int_0^\infty dr\, r\,\langle\Theta(r)\mathscr{T}(0)\rangle_n, \tag{55}$$

where $\Theta(r)$ is the energy-momentum tensor. Employing a standard form factor expansion, it is then possible to recover the conformal dimension of the twist field (3) from its two-point function with the energy-momentum tensor. This computation is possible thanks once more to the results of [39] where the form factors of the energy-momentum tensor were obtained, in particular all one- and two-particle form factors. Fig. 3 shows the numerically obtained values of $\Delta_n$ employing (55). As can be seen it is possible to compute these values also for non-integer $n$ as the form factors as well-defined for all values of $n$. It is also noticeable that the saturation of the $\Delta$-sum rule becomes worse the larger $n$ is. This is a rather common feature of the branch point twist field which is due to the fact that all form factor contributions to the expansion are proportional to $n$ and so the weight of further form factor corrections is increased as $n$ increases. We have also observed that the numerical determination of the constants $u(n)$ and $v(n)$ in the asymptotics (44) becomes more difficult for larger $n$. Both constants are the result of the numerical integration of a decaying but wildly oscillating function which is delicate. A less precise knowledge of the values $u(n)$ and $v(n)$ may well also contribute to the worsening agreement observed in Fig. 3. Fortunately though we have very good agreement for $n = 2$ which both confirms the one-particle form factors are correct and shows that including the six first contributions to the form factor expansion leads to nearly exact results.

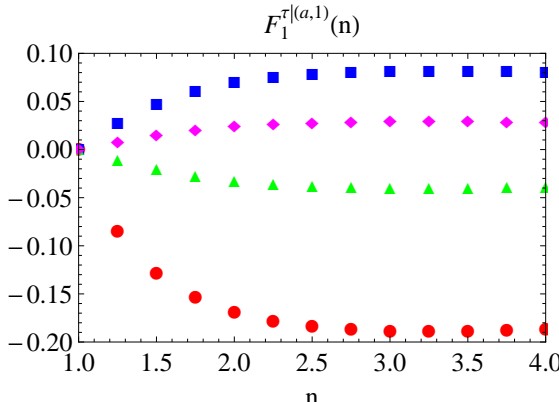

Figure 4: One particle form factors associated to particle 1 (circles), particle 2 (squares), particle 3 (triangles) and particle 4 (rombi).

The first four one-particle form factors $F_1^{\mathcal{T}|(a,1)}(n)$ for $a = 1, 2, 3, 4$ are all real numbers and their values are presented in Fig. 4. All values are rather small and, in absolute value, are smaller the higher the particle label. In particular:

$$F_1^{\mathcal{T}|(1,1)}(2) = -0.169286, \qquad F_1^{\mathcal{T}|(2,1)}(2) = 0.0687845,$$
$$F_1^{\mathcal{T}|(3,1)}(2) = -0.0336516 \quad \text{and} \quad F_1^{\mathcal{T}|(4,1)}(2) = 0.0230515. \tag{56}$$

We can also numerically determine the values of the coefficients $A_{11}^{1a}(n)$, $B_{1a}^{11}(n)$ and $C_{1a}^{11}(n)$ for $a = 1, 2$. For $n = 2$ they are:

$$A_{11}^{11}(2) = 0.0502866, \quad A_{12}^{11}(2) = -0.0387777,$$
$$B_{11}^{11}(2) = 0.0000930, \quad B_{12}^{11}(2) = 0.0002876,$$
$$C_{11}^{11}(2) = 0.0091876, \quad C_{12}^{11}(2) = -0.0043528. \tag{57}$$

With these values, it is now possible to evaluate the integrals (53)-(54):

$$\int_{-\infty}^{\infty} d\theta \, \frac{h_{11}^{11}(\theta, 0; 2) + h_{11}^{12}(\theta, 0; 2)}{\sqrt{\cosh \frac{\theta}{2}}} = 0.100857,$$
$$\int_{-\infty}^{\infty} d\theta \, \frac{h_{12}^{11}(\theta, 0; 2) + h_{12}^{12}(\theta, 0; 2)}{\sqrt[4]{1 + \frac{m_2^2}{m_1^2} + 2\frac{m_2}{m_1}\cosh \theta}} = 0.0181714. \tag{58}$$

## 3.2 Exact corrections to saturation

Putting together all the numerical values found in the previous section, we see that the formula (51) may be expressed as:

$$S_2(\ell) - S_2(\infty) = -0.0182441 K_0(m_1 \ell) - 0.00301205 K_0(m_2 \ell)$$
$$- 0.000720926 K_0(m_3 \ell) - \frac{1}{2\pi^2} \sum_{j=1}^{2} \int_{-\infty}^{\infty} d\theta \, h_{11}^{1j}(\theta, 0; 2) K_0(2m_1 \ell \cosh \theta/2)$$
$$- 0.000338282 K_0(m_4 \ell)$$
$$- \frac{1}{2\pi^2} \sum_{j=1}^{2} \int_{-\infty}^{\infty} d\theta \, h_{12}^{1j}(\theta, 0; 2) K_0(\ell \sqrt{m_1^2 + m_2^2 + 2m_1 m_2 \cosh \theta}), \tag{59}$$

where the leading contributions to the integrals are $0.00452814\frac{e^{-2m_1\ell}}{\sqrt{m_1\ell}}$ and $0.00115377\frac{e^{-(m_1+m_2)\ell}}{\sqrt{m_1\ell}}$, respectively. From (59) it is clear that the one-particle form factor contributions are led by small coefficients (as the one-particle form factors are all small numbers). However, the two-particle contributions are characterized by even smaller coefficients and are in fact strongly suppressed (they are essentially negligible for $m_1\ell$ above 1), as shown in Fig. 5. Therefore

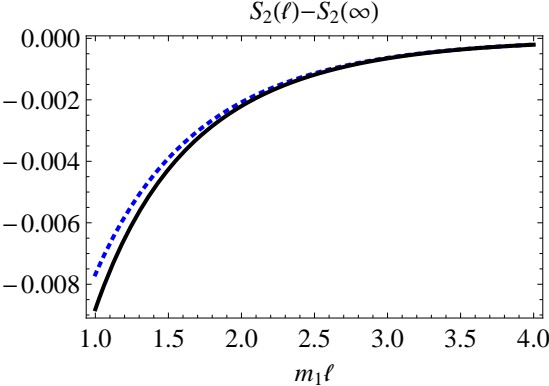

Figure 5: The function (51) (solid curve). The dashed curve is the contribution $-0.0182441K_0(m_1\ell)$, that is the leading contribution to (51). Clearly the contribution proportional to $K_0(m_1\ell)$ is leading for the full range of values of $\ell$ considered here.

the suggestion put forward in [7] that the one-particle form factors are zero and therefore not detectable in TCSA is mistaken. In the next section we will review how this suggestion was arrived at in [7] and clarify the origin of this apparent mismatch between TCSA and form factor results.

## 4 Comparison to TCSA results

A good way to gain at least a qualitative understanding of how the TCSA results and the form factor results compare is to display them in the same graph. This is what is shown in Fig. 6. More precisely, the next-to-leading order corrections to the second Rényi entropy obtained

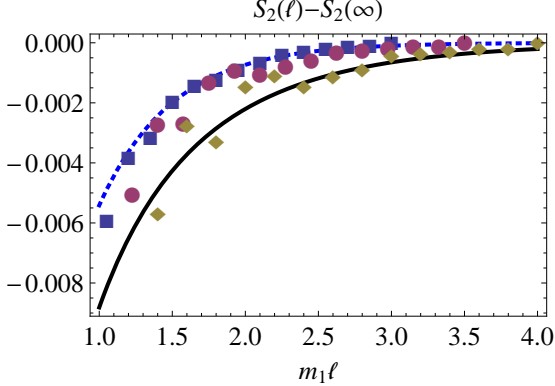

Figure 6: The function (51) (solid curve), compared to TCSA results obtained in [7] for a system of length $m_1L = 8$ (rombi), $m_1L = 7$ (circles) and $m_1L = 6$ (squares) and to the fit (60) (dashed curve).

from TCSA and from form factors are presented. Let us describe the data in some detail:

- The black solid line represents such corrections as are obtained from an exact form factor calculation which includes the first six terms of the form factor expansion as shown in equation (51). As shown in Fig. 5 out of these six contributions, the first one is very dominant, with subsequent contributions being strongly suppressed. This suggests that the solid curve in Fig. 6 provides a nearly exact description of the corrections to saturation of the entropy of a subsystem of size $m_1 \ell$ in an *infinite* system.

- The numerical data in the Fig. 6 represented by squares, circles and rombi are the TCSA results for *finite* systems of lengths $m_1 \ell = 6, 7$ and 8, respectively. These are the same data as employed in [7] but where the saturation value has been subtracted[1]. Since the TCSA approach (by construction) can only deal with finite systems, the expectation is that a match with a form factor computation will only be achieved for large system sizes (how large will typically depend on the quantity that is being computed). Observing Fig. 6 we can conclude that agreement with the form factor data is indeed better as volume increases (as expected) and is already very good for the data corresponding to $m_1 \ell = 8$.

- The final element of Fig. 6 is the dashed curve which depicts the function

$$f(\ell) = -0.04 e^{-2m_1 \ell}. \tag{60}$$

This function is the result of fitting the numerical TCSA data corresponding to $m_1 \ell = 6$ with a decaying exponential[2]. A similar fit of the $m_1 \ell = 5$ data was used in [7] to deduce the rate of decay of the exponential corrections to saturation. This fit plays an important role as it is from this *single* evidence that the author of [7] concludes there are no corrections to entanglement of the form $e^{-m_1 \ell}$ and therefore the one-particle form factors must be zero.

From Fig. 6 and the points above it would seem that two contradictory conclusions follow: the TCSA data and the form factor data agree rather well, yet the exponential decay identified in [7] disagrees with the form factor calculation. Where does this mismatch come from?

This question is easy to answer and the answer lies in the manner in which the exponential decay of the TCSA data was (wrongly) identified. This wrong identification is due to the fact that the fit (60) is a good fit of the $m_1 \ell = 6$ data but clearly not a good fit of the larger system size data. The correct manner of identifying the exponential decay from TCSA would have been to first find a reasonable infinite volume extrapolation of the data and then find a fit of that extrapolation. We would expect such a procedure to give a result much more in tune with the form factor analysis.

## 5 Conclusions

In this paper we have carried out an in depth study of the second Rényi entropy in the transverse field Ising model or $E_8$ minimal Toda field theory employing the branch point twist field approach developed in [24]. This work, follows on from a stream of works where the entanglement of various particular integrable models has been studied [24, 25, 51–54] by the same

---

[1] I would like to thank Tamás Pálmai for sharing with me these numerical data.

[2] This fit was carried out by Tamás Pálmai who shared it with me in private communications. The fit is alluded to in [7] but not given explicitly there.

method. Our results provide the fist detailed form factor study of the entanglement of an interacting theory with a complicated particle content and no internal symmetries so that all twist field form factors are non-vanishing. We have compared our results to those obtained by a new method for the evaluation of measures of entanglement developed by T. Pálmai [7]. This new method is based on the use of the truncated conformal space approach [29] to massive integrable quantum field theories.

One of the motivations for this work was to check the veracity of the claim that the leading corrections to saturation of the second Rényi entropy are not those predicted by the twist field approach. This claim was put forward in [7] based on the analysis of some TCSA data. In this paper we have shown that the leading corrections to saturation are indeed those expected from a twist field form factor analysis and that there was instead a fault in the interpretation of the TCSA data. The main conclusion about the exponential decay of entanglement had been reached based on data for relatively small system sizes and, not surprisingly, these data did not reproduce correctly the infinite size behaviour described by the form factor approach. Once this point is clarified there is in fact no contraction between the form factor and TCSA approaches and, based on the data at our disposal, we can say that they agree reasonably well already for systems sizes of the order of $m_1 \ell = 8$.

We conclude from this analysis that it is actually rather difficult to correctly identify subleading exponential corrections to entanglement purely from a TCSA analysis, as this requires the ability to extrapolate numerical data to the infinite size limit with great precision. In cases where this is possible, TCSA is a good numerical alternative to the form factor approach which, although leading to exact results, can be hard to use in complicated models. The lengthy computations needed to arrive at (51) illustrate this point rather well.

**Acknowledgments:**   I am greatly indebted to Tamás Pálmai for many extremely useful e-mail discussions regarding the TCSA approach and the sense in which his numerical results should be comparable to the form factor calculation presented here. I thank him too for constructive feedback on the paper, for sharing his numerical data with me, as presented in Fig. 6 and for even carrying out additional numerics to help me answer the main question raised in this paper. I am also grateful to Benjamin Doyon for many useful and interesting discussions, to Andreas Fring for sharing his extensive knowledge of (and notes on) the Toda $S$-matrices with me, to Patrick Dorey for bringing references [33, 34] to my attention and to Gerard Watts for answering some of my questions about the TCSA approach. Finally, I would like to thank Pasquale Calabrese for hospitality during my visit to SISSA in July of 2016. It was during this visit that the present work was initiated. I gratefully acknowledge support from EPSRC through grant EP/P006108/1 "Entanglement Measures, Twist Fields, and Partition Functions in Quantum Field Theory".

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
