# Peer review of "Massive Corrections to Entanglement in Minimal E8 Toda Field Theory"

_SciPost Physics, doi:SciPost Phys. 2, 008 (2017)_

## Round 1 · Referee Report · Anonymous (Referee 1) · 2016-12-8

Strengths

1- The paper presents a simple introduction to the IMMF model. 2- The FF analysis is clear and to my level of understanding appears correct. 3- The paper sets out to answer a question arising from a controversial conclusion of a recent publication.

Weaknesses

See general report. The main problem is in the qualitative and overly discursive nature of the comparison with TCSA results.

Report

This paper presents a form-factor (FF) expansion calculation of the the 2nd Renyi entropy of the so called IMMF model. This main stated purpose is to compare with a recent TCSA calculation of Palmai [7] that came to the surprising conclusion that the sub-leading contribution appeared to be fitted by a function exp(-2 m_1 l) as opposed to the anticipated exp(-m_1 l). The explanation of the FF expansion for the correlation function of twist fields is generally clear and follows a fairly well-established argument that draws on and extends the results of [24] and [37]. Up to the end of Section 3 the results are convincing.

The paper has weaknesses however in Sections 4 and 5. The paper is supposed to be carrying out a comparison of two approximate methods (the FF method is still an approximation due to the truncation involved) but the comparison carried out is qualitative and sometimes the conclusions are unclear. In particular:

(1) A key difference between the two techniques is that TCSA method works for finite m_1 L and the FF for infinite m_1 L. In order to compare them in any way, some attempt at extrapolation of the TCSA results to L infinite needs to be made, but this is currently lacking.

(2) Further to (1), it is stated that `it has been observed numerically that higher [m_1 L] values produce comparable results', but no data is presented to explain or back this statement. Is this a private communication from Palmai, or an independent TCSA calculation? Either way, it would be very helpful to show this higher L data and to include it in the comparison.

(3) The comparison of fits described in the penultimate paragraph of Section 4 is just qualitative. There are of course many quantitative ways of comparing fits and I think that it is reasonable to expect that such an analysis should be carried out (after carrying out the extrapolation described in (1)).

(4) The statement is made 'there are some oscillations due to numerical inaccuracy'. I'm not sure what this means and no error bars are given. Is the numerical accuracy resulting from poor numerical evaluation of truncated TCSA results, or something to do with the TCSA truncation level itself (in which case the error is
not really numerical). Also is there any evidence of the dependence of the TCSA results on the truncation level?

(5) Section 5 is again overly discursive and sometimes seems self-contradictory. For example, 'a priori the results of [7] seemed to disagree with a prediction based on branch point twist fields', 'the TCSA data are not precise enough to differentiate [the two behaviours]', 'the agreement between numerical data and the form factor data [..] is still remarkable', 'the [two approaches] complement each other and lead to consistent results'.

It is hard to extract a clear message from this discussion. Is the final conclusion simply that the L-extrapolated TCSA data fits (51) well and better than (60)? If so, this should be clearly stated clearly and ideally backed by quantitative evidence.

For the paper to be accepted, I would suggest that changes have to be made to Sections 4 and 5 so that the comparison is quantitative and clear, otherwise, in my opinion, the stated purpose of the paper is not met.

Requested changes

1- A reworking of Sections 4 & 5 to address the points raised in the general report.

  • validity: high
  • significance: good
  • originality: high
  • clarity: ok
  • formatting: good
  • grammar: good

Author:  Olalla Castro-Alvaredo  on 2017-01-20  [id 88]

(in reply to Report 1 on 2016-12-08)
Category:
answer to question
reply to objection

Answer to Referee Reports

I would like to thank both referees for their comments and critical reading of the manuscript. The main criticisms raised by both referees are pretty much the same and essentially have to do with how the main conclusions and discussion of results are presented. I agree with the referees there were clearly some shortcomings in my discussion of the results and I hope the changes I have introduced have improved this. Indeed the second referee is quite right that these too sections were too “discursive” and a more to-the-point discussion would have been clearer. The conclusions from my work are actually straightforward so it is a shame it did not seem so in the paper. I believe a fault of the original version of the paper is that I devoted too much space to discussing the TCSA results (which are not mine) rather than focusing on what I have actually done in this work. Both referees have recommended a more clear/to the point rewriting of sections 4 and 5 of the paper, devoted to discussion of results and conclusion. This has been done in the new version of the paper, as the referees will see and the one result is that both sections are considerably shorter and I hope clearer. I have also answered their queries as raised in the report. My answers follow below. I hope the changes to the paper and my answers below will be sufficient to convince the referees that the paper is of good quality and fit to be published in SciPost.

Answer to Points raised in Report 52 on 2016-12-18

  1. The author points out that the fit (60) does not agree well with the higher volume (m1L=8) TCSA data, while the agreement with the form factor result (51) improves with increasing volume. However, it is not clear what was the volume L for which the fit (60) was obtained. Since it is a numerical fit, it needs to be re-evaluated for m1L=7 and m1L=8, however, Fig 6. sugests this was not done.

The referee is right here that I forgot to say clearly which data are fitted by (60). I had actually mentioned this in the conclusion but it is clear that it would have been better to say this in context of the discussion around Fig. 6. The fit (60) is a fit of the data for m1L=6. The referee is right that it would have made sense to fit also higher volume data and indeed it is because this was not done that Palmai finally arrived at the wrong conclusion. The fit (60) is the fit that was used by Palmai in his work and from which he draws the conclusion that there is no one-particle contribution to the EE. This is the reason why only this particular curve is presented in my paper. Clearly he should have carried out further fits and comparisons before drawing such as a stark conclusion, especially as the volume was not particularly large. My understanding is that he had more confidence on his lower volume data than on data for higher volumes and decided to take m1L=6 as his benchmark rather than trying to extrapolate to infinite volume. All of this is explained more clearly now in section 4.

  1. Also, the significance of deviations between either (51) or (60) on the one hand, and the TCSA on the other, is predicated on the understanding of the truncation error inherent in TCSA. The author mentions in passing “some oscillations due to numerical inaccuracy”. While it is clear that here the author is relying on data from T. Palmai, it is unclear what this means and also what is the justification for assigning these “oscillations” to TCSA error since no estimate of truncation errors is given.

The referee is right that my statement about oscillations was obscure and I have now removed it. To clarify, there is a truncation error, which is inherent to TCSA. This error is unknown to me, as Palmai did not share it with me. I only know what he says in his paper where he writes that the numerical results are “reliable and relatively precise” despite “extremely low cut-offs when truncating the conformal space”. Beyond this, there are some general statements about TCSA that can be made without knowing the precise magnitude of the truncation error. One such statement is that if the same truncation is used for different values of the volume (in his paper Palmai states that he has used 18 basis states), then it is expected that data will become worse as volume increases. The reason for that is that the TCSA approach is perturbative in such a way that the size of the perturbation grows with the volume. This is the reason why one expects the higher volume data in Fig. 6 to get worse (my original statement was made in view of this fact). I have avoided any discussion of this point in the new version since these are not my data and a detailed discussion of the truncation error of these data would not change the results and conclusion of my paper.

  1. The author also states “The reason for this is that although m1L = 8 may not seem a large value, it has been observed numerically that higher values produce comparable results.” While it may very well be true that m1L=8 is sufficiently close to infinite volume, I do not see the evidence, and no details are given concerning the numerical observations mentioned here.

The evidence in this particular case comes both from private communications with Palmai where he stated that computations for volumes higher that 8 produced very similar results and also from the statement “... we also observe that at LM = 5 the finite volume effects are gone and the entropy function becomes independent of L. (Slight truncation effects can be seen in the form of a drift with increasing volumes)” which can be found in his paper. What he is saying here is that after volume 5 the only relevant effect that he can observe when plotting the Renyi entropy for higher volumes is a “drift” which we can see pretty clearly in my Fig. 6.

However, his additional claim that the L dependence disappears after LM=5 may be correct at the level of the leading contribution to the Renyi entropy (saturation) but is not true when looking a sub-leading corrections, which is what I am studying in my work. At the level of such corrections, there is a drift but there is also a change in the slope of the various curves. That is why his exponential fit of the m1L=6 data is not a good fit of the m1L=8 data and also why it is in fact hard to extract the right exponential decay without having a reliable infinite volume extrapolation (which is absent in his work).

For the purposes of my work I believe it is sufficient to say that the agreement with my data for m1L=8 is reasonably good which is consistent with the statement that the m1L=8 data are close to the infinite volume data.

  1. By considering the fact that the form factor calculations have a firm theoretical basis, and that it clearly agrees quite well with TCSA data (despite all the issues I mentioned above), I would have expected the author to draw a clear and unambiguous conclusion, especially since the fit (60) is much more heuristic.

The conclusions have now been written in a more unambiguous fashion, they are more to the point, much shorter and hopefully clear. The main conclusion is that the TCSA approach, while providing a good numerical estimate of the exponential corrections to saturation for different system sizes, is generally not good enough to enable a precise identification of the rate of this decay at infinite volume unless a very reliable extrapolation to infinite volume is known.

  1. The author also discusses some points in separate places, which makes it harder to put together her statement regarding the issue. For example, in Section 4:

“Therefore, prior to this study, it was not possible to know if the leading term proportional to K0(m1l) was going to be suppressed by, for instance, an extremely small value of the one-particle form factor.”

Then in Section 5 the 2nd to 4th paragraphs seem to be devoted to precisely this problem. It would be better to decide on where to discuss this point, and rewrite the relevant part of the text in a more focused way, clearly stating the conclusion and the evidence/considerations supporting it.

Section 4 now contains a more focused and organized (using bullet points) discussion. Some of the points above have been removed altogether as they did not help understanding.

  1. Examples of English grammar mistakes/misprints/badly formulated sentences:

I agree there are some typos in the paper but I disagree with the earlier statement (listed as a weakness) that they make the paper difficult to understand. In fact, the main examples the referee gives below contain no typo or very small ones. It seems to me some of the criticism has to do with expressions that are commonly used in British English but perhaps not so in American English. This does not mean they are incorrect. Having said this the major changes to sections 4 and 5 mean that most of the sentences below do not feature in the paper anymore.

p. 11 “This obviously posses the question”. There is a typo here in that it should be “poses” not “posses”. To “pose a question” means to ask a question. See e.g.: http://idioms.thefreedictionary.com/pose+a+question p. 13 “may well also contribute” (may or well? Possibility or a certainty?) There is nothing grammatically wrong with this sentence. “May well” is an English expression meaning “it is likely to”. You can find further details at: http://dictionary.cambridge.org/dictionary/english/may-well p. 16 “as volume in increased” This should be “as volume is increased”. p. 17 “wee know from” This should be “we know from” p. 17 “to be large of small” This should be “to be large or small” p. 17 “It is therefore desirable that the TCSA approach may be extended to” (may??) The use of “may” in this sentence is perfectly correct. It just means that “it could be done but it is not sure to happen”. I could not find a similar example on-line but I have checked this one up with some of my native English colleagues and they all recognize it as a correct sentence.

  1. Figures (especially axis labels) are not well formatted; some of the text eventually overlaps, due to too large fonts used. I have changed all figures where this problem occurred to make axes labels smaller and to display the name of the depicted quantities on top of the picture frame so there are no overlaps. I think it looks cleaner now.

Requested changes: 1. A revision of Sections 4&5 in order for the presentation be more focused and clear, with the conclusions stated clearly.
Sections 4 & 5 have been almost entirely rewritten. Section 4 is now organized with the help of bullet points so that the main conclusions should be clearer. 2. Clarify the issues surrounding the comparison to TCSA, stating the significance of the observed agreement/deviations more clearly. This is now done is section 4 and 5. Both sections are much shorter and contain fewer details about the TCSA data as in the previous version but I believe they contain the amount of detail needed for the paper to be understandable and the conclusions clear. Correct the grammar mistakes/misprints, especially those influencing the comprehensibility of the text. I have corrected all typos I found, including some pointed out by the referee. 3. Correct font sizes in figures, eliminate text overlaps. Done

---

## Round 1 · Referee Report · Anonymous · 2016-12-08

1. A key difference between the two techniques is that TCSA method works for finite m_1 L and the FF for infinite m_1 L. In order to compare them in any way, some attempt at extrapolation of the TCSA results to L infinite needs to be made, but this is currently lacking.

It is true it would have been helpful to have such an extrapolation but unfortunately this was not given in the work of Palmai. What was stated however (in private communications and also in his paper) is that results do not change significantly when increasing the volume beyond m1L=8. The precise statement in his paper is “... we also observe that at LM = 5 the finite volume effects are gone and the entropy function becomes independent of L. (Slight truncation effects can be seen in the form of a drift with increasing volumes)”. Although this statement refers to the full entropy function (including saturation) it says that the only observable effect of increasing the volume is a small “drift”. We can see this drift quite clearly in my Fig. 6. From these two sources I think it is reasonable to believe that the m1L=8 curve is a good approximation of the infinite volume results and of course this is confirmed by the fact that the m1L=8 data do indeed agree pretty well with the form factor data.

  1. Further to (1), it is stated that `it has been observed numerically that higher [m_1 L] values produce comparable results', but no data is presented to explain or back this statement. Is this a private communication from Palmai, or an independent TCSA calculation? Either way, it would be very helpful to show this higher L data and to include it in the comparison. As stated above this is a private communication and can be also read off from the statement I quote above, from Palmai’s paper. Palmai did not share his higher volume data with me and he is not obliged to, so all I can work with is his paper and what he has stated in our correspondence. It would have been nice to have those results but having them would not change the conclusions and main results of this paper so I do not think them essential.

  2. The comparison of fits described in the penultimate paragraph of Section 4 is just qualitative. There are of course many quantitative ways of comparing fits and I think that it is reasonable to expect that such an analysis should be carried out (after carrying out the extrapolation described in (1)).

I agree the extrapolation proposed in (1) would be desirable but this is beyond the scope of my paper and of what I can personally do since it relates to Palmai’s work, not mine. My main objective was to present a rigorous exact computation that would disprove Palmai’s claim that the one-particle form factors are zero. I have achieved this through my form factor calculation which being an infinite volume computation is no doubt more accurate that any of the data TCSA can generate. I can only compare my exact result to the TCSA data that I have been given. This may be imperfect but it is in my view enough to make the point I wanted to make, namely that a simple fit of TCSA data does not provide an accurate description of the rate of decay of corrections to the Renyi entropy (unless an infinite volume extrapolation is known). So far only a form factor calculation can do this with precision. Therefore drawing conclusions about such corrections from TCSA computations is risky and, as with Palmai’s conclusion about one-particle form factors, likely to be wrong.

  1. The statement is made 'there are some oscillations due to numerical inaccuracy'. I'm not sure what this means and no error bars are given. Is the numerical accuracy resulting from poor numerical evaluation of truncated TCSA results, or something to do with the TCSA truncation level itself (in which case the error is not really numerical). Also is there any evidence of the dependence of the TCSA results on the truncation level?

Palmai’s work does not provide an assessment of the truncation error. He only states that the data are “reliable and relatively precise” despite “extremely low cut-offs when truncating the conformal space”. I agree that my statement about numerical inaccuracy is unhelpful and I have now removed it from the paper altogether. I think a discussion of the numerical accuracy of Palmai’s data is beyond the scope of this work and would not change the results or the conclusion of the paper. The point I was trying to make with the sentence the referee quotes is that (whatever the truncation error is) the data are expected to become less reliable as volume increases. This is because of the intrinsic nature of the TCSA approach, which is such that the size of the perturbation grows with the volume. Therefore, for a constant truncation level, data will become less reliable at higher volumes.

  1. Section 5 is again overly discursive and sometimes seems self-contradictory. For example, 'a priori the results of [7] seemed to disagree with a prediction based on branch point twist fields', 'the TCSA data are not precise enough to differentiate [the two behaviours]', 'the agreement between numerical data and the form factor data [..] is still remarkable', 'the [two approaches] complement each other and lead to consistent results'.

I have organized the discussion differently so that (hopefully) the various statements do not seem contradictory anymore. The discussion is now much shorter and to the point so many of the statements above are now not in the paper.

  1. It is hard to extract a clear message from this discussion. Is the final conclusion simply that the L-extrapolated TCSA data fits (51) well and better than (60)? If so, this should be clearly stated clearly and ideally backed by quantitative evidence.

Yes, this is a correct conclusion and, one may say, not a surprising one: the higher the volume the better the agreement with infinite volume QFT. On the other hand (60) is just a fit of the data for volume m1L=6 and, as it turns out, gives a completely wrong prediction for the form of the exponential corrections at infinite volume. The only quantitative evidence I can provide is the evidence that follows from my own work with form factors and from that point of view the conclusion is absolutely clear.

  1. For the paper to be accepted, I would suggest that changes have to be made to Sections 4 and 5 so that the comparison is quantitative and clear, otherwise, in my opinion, the stated purpose of the paper is not met.

I have rewritten sections 4 and 5 so that the description of the data in Fig. 6 is now organized through bullet points, followed by a brief discussion. The final conclusion in section 5 is much shorter and to the point. I think the referee is asking for too much in terms of quantitative results. There are strong, exact quantitative results in the paper already, following from a very non-trivial form factor calculation and those are strongly backed by cross-checks such as the delta sum rule. To seek further quantitative precision on the TCSA side is to ask me to do the work Palmai did not present in his own paper and I think this is not fair or achievable. My claim that the one-particle contribution is (as expected) the leading one is independent of any TCSA data. It is just that those data and some of the (wrong) conclusions drawn from them have serve as inspiration for this work.

Other changes:

Apart from the changes already mentioned in my answers I have also made a few other corrections:

I have added references [33] and [34] which relate to the historic development of the study of Affine Toda Field Theories (I thank P. Dorey for pointing these out).

I have updated [54] which is now published.

---

## Round 1 · Referee Report · Anonymous (Referee 2) · 2016-12-18

Strengths

  1. The scaling of entanglement entropy with subsystem size in massive quantum field theories is an interesting and timely topic.
  2. The general exposition and the theoretical computation in Sections 1-3 is clear and well described, and gives a very good example of the application of the twist field form factor method to compute Renyi entropies.
  3. The results are also novel and interesting.

Weaknesses

  1. The discussion of the comparison between the form factor expansion and numerical TCSA data is quite superficial in many respects, some important points are glossed over.
  2. The author does not state sufficiently clearly her conclusion regarding the problems considered in the paper.
  3. There are some English grammar mistakes/misprints in the text, some of them also affecting its comprehensibility by the reader. There are also some formatting issues in the figures.

Report

The paper presents a form factor based calculation of the second Renyi entropy in the E8 (Ising) model. The aim is to compare it to recent TCSA results obtained by T. Palmai, and clarify the role played by one-particle contributions. The main problems with the text concern Sections 4 and 5 and correspond to items 1 and 2 in the list of weaknesses above.

  1. The author points out that the fit (60) does not agree well with the higher volume (m1L=8) TCSA data, while the agreement with the form factor result (51) improves with increasing volume. However, it is not clear what was the volume L for which the fit (60) was obtained. Since it is a numerical fit, it needs to be re-evaluated for m1L=7 and m1L=8, however, Fig 6. suggests this was not done. Also, the significance of deviations between either (51) or (60) on the one hand, and the TCSA on the other, is predicated on the understanding of the truncation error inherent in TCSA. The author mentions in passing “some oscillations due to numerical inaccuracy”. While it is clear that here the author is relying on data from T. Palmai, it is unclear what this means and also what is the justification for assigning these “oscillations” to TCSA error since no estimate of truncation errors is given. The author also states “The reason for this is that although m1L = 8 may not seem a large value, it has been observed numerically that higher values produce comparable results.” While it may very well be true that m1L=8 is sufficiently close to infinite volume, I do not see the evidence, and no details are given concerning the numerical observations mentioned here.

  2. By considering the fact that the form factor calculations have a firm theoretical basis, and that it clearly agrees quite well with TCSA data (despite all the issues I mentioned above), I would have expected the author to draw a clear and unambiguous conclusion, especially since the fit (60) is much more heuristic. The author also discusses some points in separate places, which makes it harder to put together her statement regarding the issue. For example, in Section 4: “Therefore, prior to this study, it was not possible to know if the leading term proportional to K0(m1l) was going to be suppressed by, for instance, an extremely small value of the one-particle form factor.” Then in Section 5 the 2nd to 4th paragraphs seem to be devoted to precisely this problem. It would be better to decide on where to discuss this point, and rewrite the relevant part of the text in a more focused way, clearly stating the conclusion and the evidence/considerations supporting it.

  3. Examples of English grammar mistakes/misprints/badly formulated sentences:

p. 11 “This obviously posses the question” p. 13 “may well also contribute” (may or well? Possibility or a certainty?) p. 16 “as volume in increased” p. 17 “wee know from” p. 17 “to be large of small” p. 17 “It is therefore desirable that the TCSA approach may be extended to” (may??)

  1. Figures (especially axis labels) are not well formatted; some of the text eventually overlaps, due to too large fonts used.

Requested changes

  1. A revision of Sections 4&5 in order for the presentation be more focused and clear, with the conclusions stated clearly.
  2. Clarify the issues surrounding the comparison to TCSA, stating the significance of the observed agreement/deviations more clearly.
  3. Correct the grammar mistakes/misprints, especially those influencing the comprehensibility of the text.
  4. Correct font sizes in figures, eliminate text overlaps.

---

## Round 2 · Referee Report · Anonymous (Referee 1) · 2017-2-14

Strengths

See my comments on version 1

Weaknesses

See my comments on version 1

Report

The author has addressed my earlier concerns and the conclusions are much clearer in the new version of the paper. I would be happy to now recommend publication.

Requested changes

None

---

## Round 2 · Referee Report · Anonymous (Referee 2) · 2017-2-18

Strengths

See report 1

Weaknesses

See report 1

Report

The author has addressed all the main points I raised in my previous review. i recommend the present manuscript for publication.

Requested changes

None

---

## Round 2 · List of Changes

Sections 4 and 5 of the paper have been largely rewritten to make the discussion and conclusions clearer.

---

## Editorial Decision

published